# Hierarchical Multiscale Diffuser for Extendable Long-Horizon Planning

## Abstract

This paper introduces the Hierarchical Multiscale Diffuser (HM-Diffuser), a novel approach for efficient long-horizon planning. Building on recent advances in diffusion-based planning, our method addresses the challenge of planning over horizons significantly longer than those available in the training data. We decompose the problem into two key subproblems. The first phase, Progressive Trajectory Extension (PTE), involves stitching short trajectories together to create datasets with progressively longer trajectories. In the second phase, we train the HM-Diffuser on these extended datasets, preserving computational efficiency while enhancing long-horizon planning capabilities. The hierarchical structure of the HM-Diffuser allows for subgoal generation at multiple temporal resolutions, enabling a top-down planning approach that aligns high-level, long-term goals with low-level, short-term actions. Experimental results demonstrate that the combined PTE and HM-Diffuser approach effectively generates long-horizon plans, extending far beyond the originally provided trajectories.

## 1 Introduction

The ability to envision a long future to plan optimal decisions is a fundamental ability of intelligent agents operating in highly complex and dynamic environments (Hamrick et al., 2020; Mattar & Lengyel, 2022). This capability allows agents to avoid suboptimal, short-sighted decisions by exploring future states that align with long-term goals, even when rewards are sparse (Silver et al., 2016; Hafner et al., 2019; Hansen et al., 2022). However, learning an effective world model (Ha & Schmidhuber, 2018) necessary for long-horizon planning is challenging due to the difficulty in modeling intricate and high-dimensional dynamics.

Traditional approaches to planning rely on learning the forward dynamics model that predicts the next state from the current state and action. Long-horizon planning is then achieved by iteratively applying one-step predictions in an autoregressive manner. A major limitation of this approach is the compounding of errors (Lambert et al., 2022), where minor inaccuracies accumulate over time. This leads to deviations from the intended trajectory and degraded performance as the planning horizon extends (Bachmann & Nagarajan, 2024). One way to mitigate this is by introducing a multiscale hierarchy (Sutton et al., 1999; Chung et al., 2017; Kim et al., 2019), where high-level planners perform planning on jumpy or temporally abstract states to reduce the frequency of planning steps.

The Diffuser approach (Janner et al., 2022; Ajay et al., 2022) extends Diffusion Models (Sohl-Dickstein et al., 2015; Ho et al., 2020) to planning tasks and has recently emerged as a promising paradigm in planning. Diffuser addresses the limitations of traditional autoregressive planning by removing the forward dynamics model. Instead, it generates an entire sequence simultaneously and holistically, similar to how image diffusion models generate all pixels. This approach eliminates error compounding and thus leads to accurate planning, particularly for long-horizon scenarios.

While Diffuser is highly effective for long-horizon planning, it faces notable limitations. A primary issue is that its planning horizon is restricted by the trajectory lengths present in the training data, making it challenging to model trajectories longer than those encountered during training. However, in many applications, the ability to plan beyond the sequence length directly experienced is essential. In contrast, planning with forward models can extend the horizon to previously unseen lengths by simply rolling out longer sequences, although this introduces compounding of errors over time. One possible solution is to collect longer training trajectories, but this significantly reduces practicality. For

example, for a robot to plan at a week- or month-long horizon based on visual experiences, it would require collecting videos of that length and training a Diffuser on those extended sequences—an approach that is highly impractical with the current Diffuser framework. Furthermore, even if such long trajectories were collected, it is well-established that planning performance degrades on these extended sequences (Chen et al., 2024b). Moreover, they would cover only a small fraction of the possible long-horizon planning space.

In this paper, we pose the following question: *How can we plan over horizons significantly longer than those available in the training data without suffering from compounding errors?* For example, can a robot create a week- or month-long plan using training data that contains only hour-long experiences? This is the challenge we tackle in this paper, a problem we refer to as *extendable long-horizon planning*. To address this, we introduce the Hierarchical Multiscale Diffuser (HM-Diffuser) framework. Our method tackles extendable long-horizon planning by dividing the problem into two subproblems: (1) extending the short original trajectories into longer ones through a process we call Progressive Trajectory Extension (PTE), and (2) efficiently training a diffusion planner on these extended trajectories by incorporating a hierarchical multiscale structure into the Diffuser framework.

PTE is a novel augmentation method that iteratively generates longer trajectories by stitching together previously extended trajectories over multiple rounds of extension. HM-Diffuser then trains on these extended trajectories, breaking down planning tasks across multiple temporal scales, enabling efficient training and execution even for very long horizons. To overcome the complexity of maintaining multiple separate diffuser models, we further introduce Adaptive Plan Pondering and Recursive HM-Diffuser, which uses a single diffuser to recursively handle different plan scales. Our results demonstrate the effectiveness of this approach in various long-horizon planning tasks, showcasing its potential to significantly advance efficient long-horizon decision-making.

The main contributions of this paper are as follows: (**i**) We introduce the problem of extendable long-horizon planning in Diffuser, where the task is to plan for trajectories longer than those seen during training. (**ii**) We propose the Hierarchical Multiscale Diffusion framework, which includes (**ii-a**) a novel augmentation method called Progressive Trajectory Extension (PTE) and (**ii-b**) a new planning diffuser, such as the Recursive Hierarchical Diffuser. (**iii**) We introduce new benchmarks, including the Extendable-Large & XXLarge Mazes, Extendable-Gym-MuJoCo, and Extendable-Kitchen, as previous benchmarks for Diffusers in the context of extendable long-horizon planning were not available in the community.

## 2 PRELIMINARIES

**Diffusion Models** (Sohl-Dickstein et al., 2015; Ho et al., 2020), inspired by the modeling of diffusion processes in statistical physics, are latent variable models with the following generative process: $p_\theta(\mathbf{x}_0) := \int p_\theta(\mathbf{x}_{0,M}) \mathrm{d}\mathbf{x}_{1:M} = \int p(\mathbf{x}_M) \prod_{m=1}^{M} p_\theta(\mathbf{x}_{m-1} \mid \mathbf{x}_m) \, \mathrm{d}\mathbf{x}_{1:M}$ Here, $\mathbf{x}_0$ is a datapoint and $\mathbf{x}_{1:M}$ are latent variables of the same dimensionality as $\mathbf{x}_0$. A diffusion model consists of two core processes: the reverse process and the forward process. The reverse process is defined as

$$p_\theta(\mathbf{x}_{m-1}|\mathbf{x}_m) := \mathcal{N}(\mathbf{x}_{m-1}|\boldsymbol{\mu}_\theta(\mathbf{x}_m, m), \sigma_m \mathbf{I}) . \tag{1}$$

This process transforms a noise sample $\mathbf{x}_M \sim p(\mathbf{x}_M) = \mathcal{N}(0, \mathbf{I})$ into an observation $\mathbf{x}_0$ through a sequence of denoising transitions $p_\theta(\mathbf{x}_{m-1}|\mathbf{x}_m)$ for $m = M, \dots, 1$. Conversely, the forward process defines the approximate posterior $q(\mathbf{x}_{1:M}|\mathbf{x}_0) = \prod_{m=0}^{M-1} q(\mathbf{x}_{m+1}|\mathbf{x}_m)$ via the forward transitions:

$$q(\mathbf{x}_{m+1}|\mathbf{x}_m) := \mathcal{N}(\mathbf{x}_{m+1}; \sqrt{\alpha_m}\mathbf{x}_m, (1 - \alpha_m)\mathbf{I}) . \tag{2}$$

The forward process iteratively applies this transition from $m = 0, ..., M-1$ according to a predefined variance schedule $\alpha_1, \dots, \alpha_M$ and gradually transforms the observation $\mathbf{x}_0$ into noise $\mathcal{N}(0, \mathbf{I})$ as $m \to M$ for a sufficiently large $M$. Unlike the reverse process involving learnable model parameters $\theta$, the forward process is predefined without learning parameters. Learning the parameter $\theta$ of the reverse process is done by optimizing the variational lower bound on the log likelihood $\log p_\theta(\mathbf{x}_0)$. Ho et al. (2020) demonstrated that this can be achieved by minimizing the following simple denoising objective: $\mathcal{L}(\theta) = \mathbb{E}_{\mathbf{x}_0, m, \boldsymbol{\epsilon}} \left[ \|\boldsymbol{\epsilon} - \boldsymbol{\epsilon}_\theta(\mathbf{x}_m, m)\|^2 \right]$. Specifically, this is to make $\boldsymbol{\epsilon}_\theta(\mathbf{x}_m, m)$ predict the noise $\boldsymbol{\epsilon} \sim \mathcal{N}(0, \mathbf{I})$ that was used to corrupt $\mathbf{x}_0$ into $\mathbf{x}_m = \sqrt{\bar{\alpha}_m}\mathbf{x}_0 + \sqrt{1 - \bar{\alpha}_m}\boldsymbol{\epsilon}$. Here, $\bar{\alpha}_m = \prod_{i=0}^{m} \alpha_i$.

**Planning with Diffusion.** Two major approaches to planning via Diffusion are Diffuser (Janner et al., 2022) and Decision Diffuser (Ajay et al., 2022). Diffuser employs the classifier-guided approach (Dhariwal & Nichol, 2021). It first trains a diffusion model $p_\theta(\boldsymbol{\tau})$ on offline trajectory data,

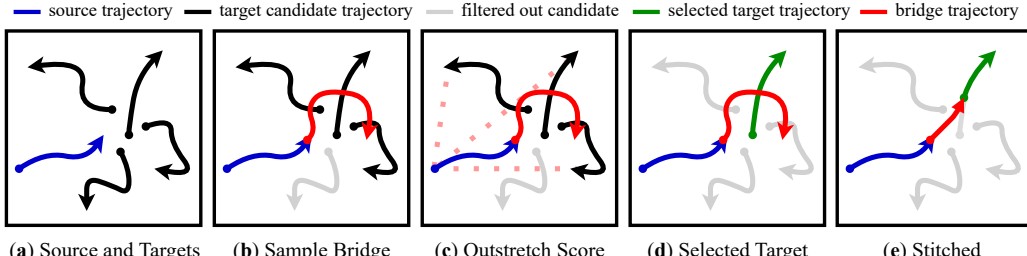

**(a)** Source and Targets    **(b)** Sample Bridge    **(c)** Outstretch Score    **(d)** Selected Target    **(e)** Stitched

Figure 1: **Progressive Trajectory Extension (PTE) (a) Source and target trajectories:** PTE starts with a source trajectory and multiple target candidate trajectories.**(b) Sampling a bridge trajectory:** A pretrained stitcher is used to roll out the source trajectory and filter out unreachable candidates. **(c) Computing the outstretch score:** For the remaining feasible candidates, an outstretch score is computed. **(d) Selecting a target trajectory:** A target trajectory is selected based on the outstretch score. **(e) Stitching:** The stitcher connects the source trajectory to the selected target trajectory, resulting in a stitched trajectory.

where each trajectory is a series of state-action pairs $\boldsymbol{\tau} = (s_0, a_0, s_1, a_1, \ldots, s_T, a_T)$. Subsequently, it trains a guidance model $p_\phi(\mathbf{y}|\boldsymbol{\tau}) \propto \exp(G_\phi(\mathbf{x}))$, with $G_\phi(\boldsymbol{\tau})$ predicting trajectory returns. This enables the construction of a modified distribution $\tilde{p}_\theta(\boldsymbol{\tau}) \propto p_\theta(\boldsymbol{\tau}) \exp(\mathcal{J}_\phi(\boldsymbol{\tau}))$. At test-time, sampling from $\tilde{p}_\theta(\boldsymbol{\tau})$ is achieved by biasing the denoising process towards $\nabla_{\boldsymbol{\tau}_m} \mathcal{J}_\phi$ of a high-return trajectory. To ensure the planned trajectory begins from the current state $\mathbf{s}$, Diffuser enforces $\mathbf{s}_0 = \mathbf{s}$ in each $\boldsymbol{\tau}_m$ during denoising. Typically, only the first action is executed before replanning from the resulting state $\mathbf{s}'$, though in simpler environments, the entire planned action sequence may be carried out. For goal-conditioned scenarios with a goal state $\mathbf{s}_g$, both $\mathbf{s}_0 = \mathbf{s}$ and $\mathbf{s}_T = \mathbf{s}_g$ are set to ensure the path terminates at the desired goal. Decision Diffuser (DD) differs from Diffuser in two key aspects: First, DD trains its diffusion model exclusively on state trajectories $\boldsymbol{\tau} = (s_0, s_1, \ldots, s_T)$, then employs an inverse dynamics model $a_t := f_\phi(s_t, s_{t+1})$ to derive actions from the completed trajectory $\boldsymbol{\tau}_0$. Second, DD implements classifier-free guidance (Ho & Salimans, 2022).

## 3 PROPOSED METHOD

Our goal is to develop a planner capable of handling planning horizons significantly longer than those in the initial dataset. Our approach consists of two phases. First, we generate longer trajectories from shorter ones using a technique called Progressive Trajectory Extension. In the second phase, we train our hierarchical multiscale planner on these extended trajectories to improve long-horizon planning.

### 3.1 PROGRESSIVE TRAJECTORY EXTENSION

The PTE process performs multiple extension rounds, progressively generating longer trajectories with each round. Before initiating a Progressive Trajectory Extension (PTE) round, we need to train a few key modules first. This includes a diffusion model $p_\theta^{\text{stitcher}}(\boldsymbol{\tau})$, referred to as the *stitcher*, which is trained using the base trajectory data $\mathcal{D}^0$. Its training process is similar to that of an unconditional diffuser (Janner et al., 2022), but adapted to operate on state sequences. We also train an inverse dynamics model $a_t = f_\theta^a(s_t, s_{t+1})$ to infer actions, and a reward prediction model $r_t = f_\theta^r(s_t, a_t)$, assuming that both can be approximated by deterministic functions.

In the $r$-th extension round, the two input datasets, $\mathcal{S}^r$ for source trajectories and $\mathcal{T}^r$ for target trajectories, and the pretrained modules are used to produce an output dataset $\mathcal{D}_{\text{out}}^r$ containing extended trajectories. Although for the first round of extension, we always have $\mathcal{S}^1 = \mathcal{T}^1 = \mathcal{D}^0$, our method offers flexibility in selecting the two input datasets $\mathcal{S}^r$ and $\mathcal{T}^r$ for $r > 1$. For instance, $\mathcal{S}^r$ can be the output of the previous round, i.e., $\mathcal{S}^r = \mathcal{D}_{\text{out}}^{r-1}$, and $\mathcal{T}^r$ as the initial dataset $\mathcal{D}^0$. For simplicity, we assume that $\mathcal{S}^r = \mathcal{D}_{\text{out}}^{r-1}$ and $\mathcal{T}^r = \mathcal{D}^0$ in the following. Within an extension round, creating a newly extended trajectory operates as follows:

*(i) Sampling source and target candidate trajectories.* We first randomly sample a source trajectory $\boldsymbol{\tau}^{\text{src}} \in \mathcal{S}^r$ along with a random batch of candidate target trajectories $\mathcal{T}_c \subset \mathcal{T}^r$. Then, we sample a state $s_t^{\text{src}}$ from $\boldsymbol{\tau}^{\text{src}}$ and a set of states $\{s_{c,t''}^{\text{cand}}\}_c$ from each candidate $\boldsymbol{\tau}_c^{\text{cand}} \in \mathcal{T}_c$.

*(ii) Sampling a bridge trajectory.* A bridge trajectory $\boldsymbol{\tau}^{\text{brg}}$ of predefined horizon length of $h$ is sampled using the stitcher with $s_t^{\text{src}}$ designated as the starting state of the bridge trajectory: $\boldsymbol{\tau}^{\text{brg}} \sim p_\theta^{\text{stitcher}}(\boldsymbol{\tau}|s_0 = s_t^{\text{src}})$. The target trajectory $\boldsymbol{\tau}^{\text{tgt}}$ is then randomly selected from a batch of candidates $\mathcal{T}_{c,\delta} \subset \mathcal{T}_c$, consisting of trajectories whose closest distance to any state in the bridge trajectory is within a threshold $\delta$. Suppose that the state $s_{t'}^{\text{brg}}$ from $\boldsymbol{\tau}^{\text{brg}}$ has the smallest distance to $s_{t''}^{\text{tgt}}$ from $\boldsymbol{\tau}^{\text{tgt}}$, then we say the stepwise distance, denoted as $k$, between $s_t^{\text{src}}$ and the target trajctory $\boldsymbol{\tau}^{\text{tgt}}$ is the number of time steps between $s_t^{\text{src}}$ and $s_{t'}^{\text{brg}}$. To finalize the bridge trajectory, we refine the bridge trajectory by resampling the trajectory form the stitcher conditioned on $s_t^{\text{src}}$ and the goal states from $\boldsymbol{\tau}^{\text{tgt}}$: $\boldsymbol{\tau}^{\text{rebrg}} \sim p_\theta^{\text{stitcher}}(\boldsymbol{\tau}|s_0 = s_t^{\text{src}}, \cdots, s_k = s_{t''}^{\text{tgt}}, \cdots, s_h = s_{t''+h-k}^{\text{tgt}})$.

*(iii) Stitching all.* This yields a new extended trajectory: $\boldsymbol{\tau}^{\text{new}} = [\boldsymbol{\tau}_{1:t-1}^{\text{src}}, \boldsymbol{\tau}_{0,t'}^{\text{rebrg}}, \boldsymbol{\tau}_{t''+1:T}^{\text{tgt}}]$. Here, square brackets denote concatenation. By adding the extended trajectory $\boldsymbol{\tau}^{\text{new}}$ to $\mathcal{D}_{\text{out}}^r$, we complete a process of generating a new extended trajectory. This process repeats until $\mathcal{D}_{\text{out}}^r$ contains the specified number of total transitions, and then for a specified number of rounds. Consequently, we obtain progressively longer trajectories as we apply more rounds.

Existing stitching methods often result in two major limitations. First, these methods frequently produce short or similarly-lengthened trajectories, with longer trajectories generated only by chance. Second, even when longer trajectories are generated, the path often loops back to the source or exhibits significant overlap. To address these issues, we introduce the following two methods.

**Tail-to-head stitching** uses the intuitive approach that trajectory extension is most effective when stitching the end of the source trajectory to the beginning of the target trajectory. To implement this, we divide the trajectory into non-overlapping segments and assign probabilities to each using a categorical distribution, as outlined in Algorithm A.1. State sampling involves selecting a segment based on the probabilities and then uniformly sampling a state within that segment. This method is simple yet flexible. For tail-to-head stitching, we assign higher probabilities to the tail of the source trajectory when sampling $s_t^{\text{src}} \in \boldsymbol{\tau}^{\text{src}}$ and to the head of the target trajectory when sampling $s_{c,t''}^{\text{cand}} \in \boldsymbol{\tau}_c^{\text{cand}}$. Setting uniform probabilities replicates standard stitching behavior.

**Outstretching** is introduced to prevent the extended trajectory from looping back to the source. This is achieved by selecting a candidate from $\mathcal{T}_{c,\delta}$ based on the top-$K$ *outstretch score*: The outstretch score is defined as the Euclidean distance between the two endpoints—the initial state of the source and the final state of the target—divided by the step distance, which approximates the actual number of steps taken in the result extended trajectory. Consequently, trajectories that loop back will have a low outstretch score, while those that extend in a straight, outward direction will have a higher score.

**Linear and Exponential PTE.** As discussed earlier, the PTE method allows for flexible input datasets for sampling source and target trajectories. This flexibility enables different types of trajectory extensions based on the dataset used. Here, we introduce two approaches. First, *Linear PTE*, the base method, where we set $\mathcal{S}_r = \mathcal{D}_{\text{out}}^{r-1}$ and $\mathcal{T}^r = \mathcal{D}^0$. As shown in Figure 3, the length of the extended trajectories increases linearly with each round. Linear PTE is a simple yet powerful extension approach that can be applied generally. However, due to its nature, it may require multiple rounds of stitching for large environments. For this reason, we introduce another PTE variant, *Exponential PTE*, where both $\mathcal{S}^r = \mathcal{T}^r = \cup_{r'=0}^{r-1} D_{\text{out}}^{r'}$. As shown in Figure 3 and Table A.4, Exponential PTE effectively extends the source trajectory, where the maximum trajectory length increases exponentially with each round. Refer to Appendix A.2 for more details.

### 3.2 Hierarchical Multiscale Diffusers

After $R$ rounds of trajectory stitching, we obtain a series of datasets, where the average trajectory length increases with each subsequent round. We then merge these into a single dataset $\mathcal{D}$ containing trajectories of various lengths. A straightforward approach would be to train a standard Diffuser (Janner et al., 2022; Ajay et al., 2022) on this dataset. However, because the dataset now includes very long trajectories, the output dimensionality $\mathcal{D}$ of the Diffuser model must scale to accommodate the longest trajectories. In real-world AI agent scenarios, this could involve a very long sequence like week- or month-long video sequences, introducing significant computational challenges. In fact, a recent study (Chen et al., 2024b) has shown that performance tends to degrade with longer horizons.

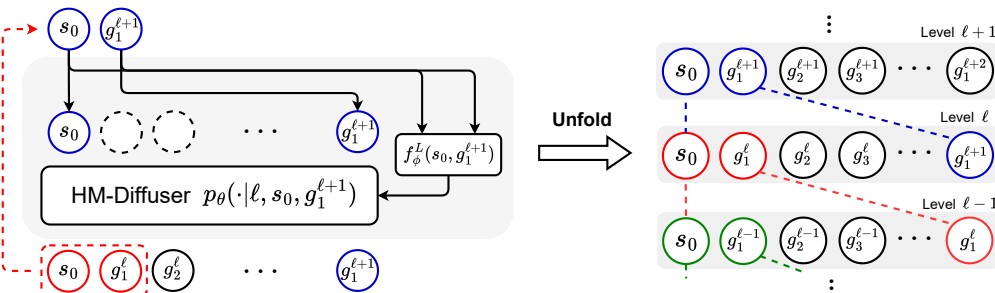

**Figure 2: Hierarchical Multiscale Diffuser (HMD)** utilizes the same model at each level, allowing for efficient multiscale planning. Assisted by the level classifier $f_\phi^L$, HMD determines the appropriate resolution of subgoals. These subgoals are recursively fed back into the model until the entire trajectory is planned.

To address this issue, we observe that the Hierarchical Diffuser (HD) approach (Chen et al., 2024b) is well-suited to our setting. Therefore, our first strategy is to apply this approach to our extended dataset generated by the PTE process. Specifically, our planner consists of a hierarchy of $L$ level-planners, $p_{\theta_\ell}(\boldsymbol{\tau})$ for $\ell = 1, \ldots, L$. The $\ell$-th planner is defined by its *jump length* $j_\ell$ and *jump count* $k_\ell$. That is, planner $p_{\theta_\ell}$ is trained on trajectories of length $H_\ell = j_\ell \times k_\ell$, randomly selected from $\mathcal{D}$. However, instead of densely utilizing all the states in the trajectory, it only considers every $j_\ell$-th state over $k_\ell$ iterations. This sparse approximation of a trajectory allows the planner to have low output dimensions for efficient computation while still mataining an effective receptive horizon of $H_\ell$. We refer to these intermediate states as *subgoals*, $g_1^\ell, \ldots, g_{k_\ell}^\ell$. The jump length at the lowest level $j_1$ is set to 1 to produce a short dense plan. The key idea of hiearchical planning is to use the first subgoal of level $\ell + 1$ to the last goal of the lower level $\ell$ for $\ell = L - 1, \ldots, 1$. That is, given the current state $s_0$, we have the following plan: $s_0, g_1^\ell, g_2^\ell, \ldots, g_{k_\ell-1}^\ell, g_1^{\ell+1} \sim p_{\theta_\ell}(\boldsymbol{\tau} | g_0^\ell = s_0, g_{k_\ell}^\ell = g_1^{\ell+1})$. We can make this condition satisfied by setting $\bar{H}_\ell = j_{\ell+1}$. That is, one jump segment of the above layer is decomposed into $k_\ell$ subgoals in the lower layer.

**Adaptive Plan Pondering.** While effective in leveraging the hierarchical multiscale structure in planning, the above approach comes with a couple of limitations. The first is the fact that the planning always starts from the highest level $L$ and goes down level-by-level to obtain the action to execute finally. It becomes an issue if the final goal is placed much nearer than the highest plan horizon $H_L$, because it would generate a long detour trajectory to move to the nearby state. To resolve this, we introduce *Adaptive Plan Pondering* (APP) by training a pondering depth predictor $\bar{\ell} = f_\phi^L(s_0, s_g)$. This is straightforward because we know the associated level of each trajectory in $\mathcal{D}$ during training. At test time, it becomes possible to start the planning directly from a lower level, when necessary, while skipping higher levels. This prevents planning inaccurate detouring and saves computation.

**Recursive HM-Diffuser.** The second inefficiency in the hierarchical multiscale diffuser described above is the need to maintain multiple diffuser models $p_{\theta_\ell}$, each with separate parameters $\theta_1, \ldots, \theta_L$. An interesting question, therefore, arises: can we use a single diffusion model to cover all levels of the hierarchy? While this may not necessarily improve performance compared to the non-shared version, which has a larger number of parameters, it would significantly reduce the complexity of managing multiple models. Therefore, it becomes a desirable approach, as long as comparable performance can be maintained. To address this, we extend the model to recursive hierarchical multiscale planning, allowing for a single diffusion model to handle the entire hierarchical structure.

We first replace the level-Diffusers, $p_{\theta_1}, \ldots, p_{\theta_L}$, by a single level-conditioned diffusion model $p_\theta(\boldsymbol{\tau} | \ell)$. Since this model must support planning across all levels, we set the output dimension of the diffuser to $\bar{d} = \max d_\ell$, where $d_\ell$ is the output dimension of the $\ell$-th diffuser (i.e., $d_\ell = (k_\ell + 1) \times \dim(s_t)$). If the required output dimension is smaller than $\bar{d}$, we mask the extra dimensions. During training, we randomly sample $\ell \sim \text{uniform}(1, \ldots, L)$ and train the parameter-shared diffuser. For planning, we predict the starting level using an adaptive plan pondering mechanism and initiate planning from that level. After obtaining a sequence of subgoals, the first subgoal is fed back into the diffuser by setting it as the final goal while decreasing the level indicator by one: $p_\theta(\boldsymbol{\tau} | \ell, g_0^\ell = s_0, g_{k_\ell}^\ell = g_1^{\ell+1})$. Repeating this process implements a form of recursive planning, where the plan is refined through cyclic iterations of a single diffuser.

## 4 RELATED WORKS

**Hierarchical Planning.** Hierarchical frameworks are widely used in reinforcement learning (RL) to tackle long-horizon tasks with sparse rewards. Two main approaches exist: sequential and parallel planning. Sequential methods use temporal generative models, or world models (Ha & Schmidhuber, 2018; Hafner et al., 2019), to forecast future states based on past data (Li et al., 2022; Hafner et al., 2022; Hu et al., 2023; Zhu et al., 2023a). Parallel planning, driven by diffusion probabilistic models (Janner et al., 2022; Ajay et al., 2022), predicts all future states at once, reducing compounding errors. This has combined with hierarchical structures, creating efficient planners that train subgoal setters and achievers (Li et al., 2023; Kaiser et al., 2019; Dong et al., 2024; Chen et al., 2024a).

**Diffusion-based Planners in Offline RL.** Diffusion models are powerful generative models that frame data generation as an iterative denoising process (Ho et al., 2020; Song et al., 2020). They were first introduced in reinforcement learning as planners by Janner et al. (2022), utilizing their sequence modeling capabilities. Subsequent work (Ajay et al., 2022; Liang et al., 2023; Rigter et al., 2023) has shown promising results in offline-RL tasks. Diffusion models have also been explored as policy networks to model highly multi-modal behavior policies (Wang et al., 2023; Kang et al., 2024). Recent advancements have extended these models to hierarchical architectures (Wenhao Li, 2023; Chen et al., 2024b; Dong et al., 2024; Chen et al., 2024a), proving effective for long-horizon planning. Our method builds on this by not only using diffusion models for extremely long planning horizons but also exploring the stitching of very short trajectories with diffusion models.

**Data Augmentation in RL** has been a crucial strategy for improving generalization in offline RL. Previous work has used dynamic models to stitch nearby states from trajectories (Char et al., 2021), generate new transitions (Hepburn & Montana, 2022), or create entire trajectories from sampled initial states (Zhou et al., 2023; Lyu et al., 2022; Wang et al., 2021; Zhang et al., 2023). More recently, diffusion models have been applied for augmentation (Zhu et al., 2023b). Lu et al. (2023) used diffusion models to capture the joint distribution of transition tuples, while He et al. (2024) extended this to multi-task settings. Li et al. (2024) used diffusion to connect trajectories through inpainting.

## 5 EXPERIMENTS

We aim to answer these questions: (1) Can HMD generate plausible trajectories significantly longer than those in the training dataset using progressive trajectory extension (PTE)? (2) Can it create feasible plans for tasks requiring much longer planning horizons than those seen in training? (3) Is our framework still beneficial when long planning horizon is unnecessary? (4) Does it remain effective in high-dimensional manipulation tasks? To facilitate our analysis, we introduce the Plan Extendable Trajectory Suite (PETS), featuring tasks from Maze2D, Gym-MuJoCo, and FrankaKitchen.

### 5.1 ANALYSIS ON THE PROGRESSIVE TRAJECTORY EXTENSION

To address our first question, we conduct illustrative experiments in the Maze2D environment. We tested the effectiveness of our proposed Progressive Trajectory Extension (PTE) process for long-horizon stitching in larger mazes. Specifically, we used the Large Maze from D4RL and designed a new XXLarge Maze (Figure A.6), which we refer to as the Extendable Maze2D benchmark.

**Datasets.** Since the existing benchmarks do not suit our problem setting, which assumes the target task cannot be solved using only the short base training data, we created base short trajectories for our maze benchmark. We began by dividing the maze into subregions of roughly equal size and defining start and goal locations for each subregion. For data collection, we randomly selected a start-goal pair within the same region and used a PD controller to collect data, navigating from start state to the goal state. Following D4RL (Fu et al., 2020), we collected 1 million transitions for each Maze setting, as depicted in Figure A.6.

**Linear PTE and Exponential PTE.** As discussed earlier, being a flexible trajectory extension mechanism, depeding on the input dataset, we can extend the trajectory either linearly or exponentially. We applied both extention strategies on the collected short base trajectories. The linear PTE method, as shown in Figure 3, gradually increases trajectory lengths, making it suitable for more stable trajectory extension. However, it may be less efficient in scenarios requiring long-horizon planning, such as in the Large and XXLarge mazes. Conversely, the Exponential PTE rapidly extends trajectory

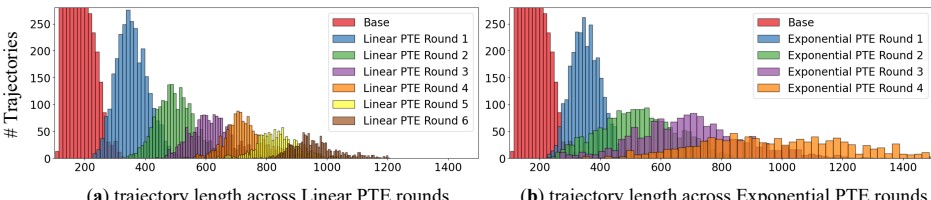

**(a)** trajectory length across Linear PTE rounds      **(b)** trajectory length across Exponential PTE rounds

**Figure 3: Trajectory Length Distribution After PTE Rounds on XXLarge Maze.** For each round of extension, the total number of transitions is restricted to 1M steps. **Left: Linear PTE** extends trajectory length at a consistent pace, with the maximum length increasing linearly across rounds. **Right: Exponential PTE** rapidly increases trajectory length, generating significantly longer trajectories by Round 4.

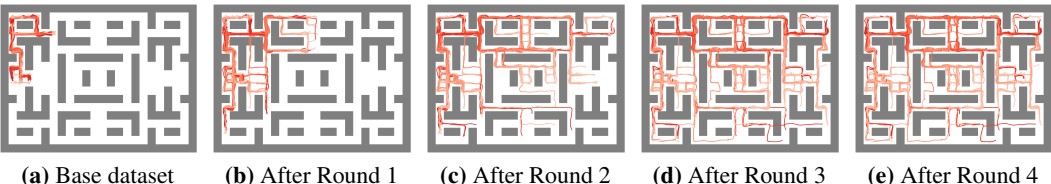

**(a)** Base dataset    **(b)** After Round 1    **(c)** After Round 2    **(d)** After Round 3    **(e)** After Round 4

**Figure 4: The Exponential PTE** method significantly enhances trajectory length. This figure visualizes trajectories that pass through the top-left corner of the maze. By the third round of Exponential PTE, the dataset has extended to cover nearly the entire maze, demonstrating the efficacy of our approach in extending trajectories.

lengths, as seen in both Figure 3 and Table A.4, offering an effective solution for managing longer trajectories. Figure 4 provides a progressive view from each round of the Exponential PTE. We can see that, starting from the top-left corner of the maze, the extended trajectories nearly spans the entire XXLarge maze more rapidly only after third rounds of extension.

## 5.2 LONG-HORIZON PLANNING

We now address our second question: Can our hierarchical multiscale planner develop long-horizon planning capabilities from these extended trajectories?

### 5.2.1 HM-DIFFUSER ON EXTENDABLE MAZE2D

**Datasets.** As the exponential PTE show efficient extension ability for long-horizon planning setting, to collect long-horion extended dataset, we applied 3 round of exponential PTE on the Large Maze base dataset and 4 round of exponential PTE on the XXLarge Maze base dataset. Subsequently, both our proposed hierarchical multiscale diffuser (HM-Diffuser) and the baseline models were trained using these datasets. Following Diffuser, we evaluated performance in two settings: (1) a single-task setting (Maze2D), where the goal was fixed and the start was randomized, and (2) a multi-task setting (Multi2D), where both the start and goal were randomized.

**Baselines.** We evaluate HM-Diffuser in comparison with Decision Diffuser (DD) and Hierarchical Diffuser (HD) across multiple planning horizons ($H = 300, 500, 1000$). The planning horizon the chosen according to number of steps required for an optimal plan to navigate between two most distant states. For instance, navigating the two farthest points in the Large Maze takes about $500$ steps, and in the XXLarge Maze, it takes $1000$ steps. Following the evaluation protocol in Diffuser, the PD controller is used during evaluation. However, to make our result more dependent on the plan instead of the PD policy, we restricted the use of the PD controller once the agent failed to reach the goal state within a specified threshold $\sigma$ after $H$ steps.

As indicated in Table 1, HM-Diffuser consistently outperformed both DD and HD across all tasks. On the single-task, Large Maze setting, this advantage was particularly noticeable at $H$=500, where HM-Diffuser scored $94.1$, significantly ahead of DD and HD, which scored $14.3$ and $28.2$, respectively. HM-Diffuser maintained robust performance even as the planning horizon increased. In the XXLarge Maze2D environment at $H$=500, it scored $47.2$, surpassing DD and HD, which scored $25.4$ and $23.2$

**Table 1: Maze2D Performance**. We compared the performance of DD, HD, and HMD across multiple horizon lengths. In every case, HMD demonstrated superior performance. Furthermore, HMD maintained consistent performance across different horizon lengths, highlighting its robustness. In contrast, both DD and HD experienced significant declines in performance as the horizon lengths increased.

| Environment | w/o PTE | w/ PTE | | | | | | | | |
|---|---|---|---|---|---|---|---|---|---|---|
| | H=100 | H=300 | | | H=500 | | | H=1000 | | |
| | DD | DD | HD | HMD | DD | HD | HMD | DD | HD | HMD |
| Maze2D-Large | $40.1 \pm 7.5$ | $20.7 \pm 4.5$ | $42.8 \pm 6.1$ | $\mathbf{104.1 \pm 8.9}$ | $14.3 \pm 2.3$ | $28.2 \pm 3.8$ | $\mathbf{94.1 \pm 9.0}$ | N/A | N/A | N/A |
| Maze2D-XXLarge | $27.9 \pm 9.2$ | N/A | N/A | N/A | $25.4 \pm 6.8$ | $23.2 \pm 7.0$ | $\mathbf{47.2 \pm 10.5}$ | $0.4 \pm 0.5$ | $3.8 \pm 1.8$ | $\mathbf{57.8 \pm 11.6}$ |
| Multi2D-Large | $31.1 \pm 7.1$ | $18.2 \pm 4.4$ | $26.6 \pm 5.2$ | $\mathbf{35.5 \pm 6.7}$ | $9.1 \pm 2.4$ | $14.2 \pm 3.1$ | $\mathbf{33.2 \pm 6.5}$ | N/A | N/A | N/A |
| Multi2D-XXLarge | $16.3 \pm 7.6$ | N/A | N/A | N/A | $10.1 \pm 4.3$ | $21.1 \pm 6.5$ | $\mathbf{38.3 \pm 9.2}$ | $3.9 \pm 1.7$ | $1.1 \pm 1.3$ | $\mathbf{31.7 \pm 8.9}$ |

respectively. At $H$=1000, HM-Diffuser continued to excel with a score of $57.8$, while DD and HD nearly failed. These results confirm that our proposed PTE framework and Hiearchical Multiscale Diffusers can effectively plan over substantially longer horizons than those seen during training.

## 5.3 OFFLINE REINFORCEMENT LEARNING

Having demonstrated efficiency and effectiveness on the Extendable-Maze2D tasks, it would be desirable for our proposed framework to also be beneficial in tasks where long-horizon planning is not necessary. Consider a scenario where we have only trajectory snippets, from which solving the target task is nearly impossible. To answer this, we will evaluate the performance of the HMD on the Extendable-Gym-MuJoCo tasks and Extendable-Kitchen tasks in this subsection.

### 5.3.1 HM-DIFFUSER ON EXTENDABLE GYM-MUJOCO

**Datasets.** Since the original D4RL (Fu et al., 2020) dataset does not align with our problem setting, we introduce a new task called Extendable-Gym-MuJoCo. This benchmark provides only the short base trajectories, which are obtained from the original D4RL offline dataset. Specifically, the original D4RL Gym-MuJoCo offline dataset was split into fixed-length segments. Considering the short-horizon nature of this task, we choose a length of $50$. The combined short and extended trajectories form the training dataset for our hierarchical multiscale diffuser and baseline models.

**Baselines.** We conducted two experimental settings to evaluate our approach on the D4RL Gym-MuJoCo benchmark as shown in Table 2. Initially, we trained the Decision Diffuser (DD) and Hierarchical Diffuser (HD) on short trajectories without progressive trajectory extension (w/o PTE) to establish baseline performances. The HD model benefits from a larger receptive field provided by hierarchical planning, whereas the DD model is limited to a flat planning structure. In our second experimental setting, to test our progressive trajectory extension (w/ PTE) process, we trained DD and HM-Diffuser (HMD) without recursion on an extended dataset. Additionally, to evaluate the effectiveness of our proposed recursive HMD, we also conducted experiments on HMD with recursion on the same extended dataset. For HMD without recursion, separate diffusion planners were trained for each level, whereas the recursive HMD variant employed a single-level conditioned diffusion model for hierarchical multiscale planning. Planning horizons were set at $H$=50 for short segments, and extended to $H$=100 for longer trajectories in the w/ PTE setting to capture a wider receptive field.

HM-Diffuser achieves the best overall performance compared to Decision Diffuser in the w/ PTE setting, as shown in Table 2. This improvement can be attributed to the hierarchical structure of HM-Diffuser, which provides a larger receptive field, facilitating more effective planning. Additionally, the recursive HMD achieves our goal by providing comparable performance to the more parameter-rich HMD-without-recursion model while it uses a single small-and-shared parameter model, reducing the burden on memory and managing multiple models. Furthermore, we observed that models trained in the w/ PTE setting generally surpass those from the w/o PTE setting, indicating the effectivness of the progressive trajectory extension machanism. As demonstrated in Figure 5, the PTE process effectively transforms trajectories with low returns into those with higher returns. We provide more investigation on PTE in the subsequent section on a high-dimensional manipulation task.

### 5.3.2 HM-DIFFUSER ON EXTEDABLE KITCHEN

High-dimensional manipulation tasks present a distinct challenge for offline reinforcement learning, where the long-horizon planning is not a necessity. To investigate how our proposed framework performs in this domain, we conduct experiments on the Extendable Kitchen task.

**Table 2: Performance on Offline Reinforcement Learning: Gym-MuJoCo**. HM-Diffuser achieves the best overall performance compared to Decision Diffuser. The results are averaged over 15 random planning seeds. Following Kostrikov et al. (2022), we emphasize in bold scores within 5% of the maximum per task.

| Dataset | Environment | w/o PTE | | w/ PTE | | |
|---|---|---|---|---|---|---|
| | | DD | HD | DD | HMD w/o Recursion | Recursive HMD |
| Medium-Expert | Halfcheetah | $68.4 \pm 1.5$ | $75.7 \pm 6.1$ | $64.0 \pm 8.2$ | $\mathbf{82.3 \pm 4.2}$ | $73.3 \pm 6.2$ |
| Medium-Expert | Hopper | $38.4 \pm 0.4$ | $81.9 \pm 8.2$ | $83.3 \pm 8.2$ | $\mathbf{94.2 \pm 6.7}$ | $\mathbf{94.2 \pm 6.4}$ |
| Medium-Expert | Walker2d | $74.7 \pm 1.9$ | $\mathbf{86.2 \pm 5.5}$ | $62.5 \pm 1.3$ | $\mathbf{83.0 \pm 1.8}$ | $71.6 \pm 2.5$ |
| **Medium-Expert Average** | | 60.5 | 81.3 | 69.9 | **86.5** | 79.6 |
| Medium | Halfcheetah | $38.2 \pm 1.6$ | $\mathbf{45.7 \pm 0.5}$ | $44.9 \pm 0.2$ | $\mathbf{45.2 \pm 0.4}$ | $\mathbf{44.8 \pm 0.4}$ |
| Medium | Hopper | $40.0 \pm 6.0$ | $52.9 \pm 2.3$ | $61.8 \pm 4.6$ | $\mathbf{87.1 \pm 1.4}$ | $82.5 \pm 0.5$ |
| Medium | Walker2d | $\mathbf{70.8 \pm 0.4}$ | $68.5 \pm 5.1$ | $58.3 \pm 6.6$ | $\mathbf{74.1 \pm 4.9}$ | $\mathbf{73.3 \pm 2.8}$ |
| **Medium Average** | | 49.7 | 55.7 | 55.0 | **68.8** | 66.9 |
| Medium-Replay | Halfcheetah | $31.3 \pm 1.6$ | $\mathbf{44.0 \pm 0.2}$ | $37.4 \pm 0.6$ | $39.8 \pm 0.4$ | $40.1 \pm 0.3$ |
| Medium-Replay | Hopper | $30.8 \pm 1.8$ | $48.0 \pm 4.2$ | $\mathbf{73.6 \pm 6.5}$ | $64.2 \pm 4.1$ | $\mathbf{70.5 \pm 5.7}$ |
| Medium-Replay | Walker2d | $16.0 \pm 0.4$ | $57.7 \pm 5.1$ | $51.4 \pm 5.6$ | $\mathbf{64.8 \pm 5.0}$ | $63.7 \pm 3.6$ |
| **Medium-Replay Average** | | 26.0 | 49.9 | 54.1 | **56.3** | 58.1 |
| **Overall Average** | | 45.4 | 62.3 | 59.7 | **70.5** | 68.2 |

**Dataset.** Similar to the Gym-MuJoCo task, to obtain our extendable kitchen benchmark, the original D4RL FrankaKitchen offline dataset was split into segments of fixed-length. Considering each subtask can be completed within a shorter horizon, we used a segment length of 20. As Table 3 illustrates, solving a subtask within this limit is very difficult (e.g., DD with No PTE). To increase observation of subtask completions, we applied three rounds of progressive trajectory extension (PTE) to the base short trajectories. We hypothesized that performance would improve with additional PTE rounds, until noise from the generated data potentially degrades performance. The final PTE round extends trajectory lengths beyond 80.

**Baselines.** We focus on assessing the efficacy of PTE and our recursive HM-Diffuser (HMD). We thus compare the performance of Decision Diffuser (DD) and HMD on each PTE round dataset. For a fair comparison, we set the planning horizon to 40 for both models on the dataset with PTE process.

To start with, DD was applied to trajectories without PTE, confirming our assumption that solving subtasks within these short trajectories is very difficult, as shown in Table 3. Following one round of PTE, DD's performance on the kitchen-partial-v0 dataset improved, averaging 2.13 subtask completions per episode. HMD showed similar results but outperformed DD on the kitchen-mix-v0 dataset, scoring 2.06 compared to DD's 0.65. After a second round of PTE, both models saw further improvements: HMD reached 2.67, surpassing DD's 2.53 on the kitchen-partial-v0 dataset and 2.53 vs. 2.50 on the kitchen-mixed-v0 dataset. HMD's superior performance is likely due to its hierarchical structure, which provides a larger receptive field. Following the third PTE round, HMD's score on the kitchen-partial-v0 task increased further to 2.73, while DD's score dropped to 2.33. On the kitchen-mix-v0 dataset, the performance of both models declined from the previous round, possibly due to some inefficiency accumulated over the PTE rounds—a topic we leave for future investigation.

**Table 3: Kitchen Task.** HM-Diffuser achieves the best overall performance among compared with Decision Diffuser. The results are averaged over 30 random planning seeds. We emphaisze the highest scores in bold.

| Task | No PTE | Round-1 PTE | | Round-2 PTE | | Round-3 PTE | |
|---|---|---|---|---|---|---|---|
| | DD | DD | HMD | DD | HMD | DD | HMD |
| Partial-v0 | $0.57 \pm 0.11$ | $\mathbf{2.13 \pm 0.27}$ | $\mathbf{2.13 \pm 0.20}$ | $2.53 \pm 0.13$ | $\mathbf{2.67 \pm 0.15}$ | $2.33 \pm 0.27$ | $\mathbf{2.73 \pm 0.11}$ |
| Mixed-v0 | $0.27 \pm 0.05$ | $0.65 \pm 0.17$ | $\mathbf{2.06 \pm 0.14}$ | $2.50 \pm 0.13$ | $\mathbf{2.53 \pm 0.14}$ | $1.50 \pm 0.17$ | $\mathbf{2.37 \pm 0.08}$ |

### 5.3.3 MORE ANALYSIS

To explore the improvements from each round of progressive trajectory extension (PTE), we analyzed the dataset obtained after each stitching round. For the Kitchen tasks with sparse reward, we focused on measuring the number of completed subtasks. To accurately count these subtask completions without duplications, we feed each state from the stitched trajectories into the true environment, which signals the completion of a valid subtask. We recoreded the number of subtasks completed

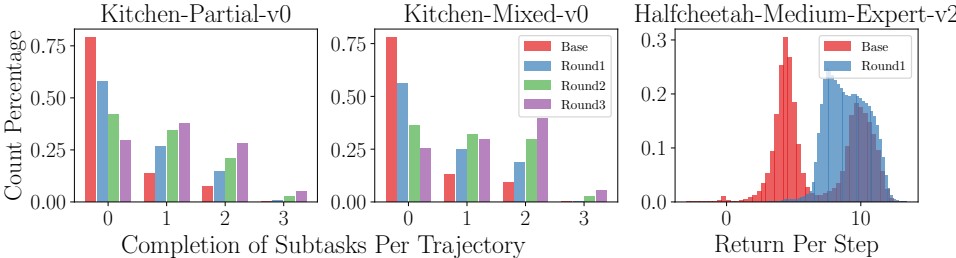

**Figure 5: Analysis of progressive trajectory extension process.** With additional PTE rounds, the subtasks completion rate increases, allowing the planner to observe more successful examples of subtasks. In the Gym-MuJoCo task, we observed a noticeable shift towards high values in the distribution of returns per step after one PTE round, indicating that trajectories initially yielding low returns evolved into trajectories with higher returns.

per trajectory. As illustrated in Figure 5, on both of the kitchen-partial-v0 and the kitchen-mixed-v0 tasks, the number of trajectories with at least one subtask completion increased with each subsequent round of stitching. Similar trends were noted for trajectories completing two and three subtasks.

For the Gym-MuJoCo tasks, where returns cumulatively increase with trajectory length, we measured the average return per step. There can be observed that a shift in the return per step distribution toward higher values, indicating that trajectories with low returns transformed into higher-return during the PTE process. For the analyses of other Gym-MuJoCo tasks, please refer to Appendix B.

## 6 CONCLUSION AND LIMITATIONS

In this work, we introduce the hierarchical multiscale diffuser framework for extendable long-horizon planning via Diffusion. Starting from a set of short trajectories that are insufficient for solving the target task, our method first extends these trajectories using Progressive Trajectory Extension (PTE). We then train a Hierarchical Multiscale Diffuser planner on this augmented dataset. In experiments, we demontrate promising results on the long-horizon Maze2D task, as well as the dense-reward Gym-MuJoCo and high-dimension manipulation Kitchen tasks.

Despite this success, our method has several areas for improvement. First, using a generative model as a stitcher limits the quality of stitched trajectories to the offline dataset used for training. Similarly, as an offline method, the effectiveness of our planner depends on the quality of the stitched dataset. Extending the approach to online fine-tuning is an important future direction. Second, the recursive version of HMD slightly underperforms compared to the non-shared HMD, likely due to differences in the number of parameters. Finding ways to enhance the shared version would be a valuable avenue for exploration. Third, our plan pondering currently predicts discrete plan levels; allowing it to regress continuous levels could improve model flexibility. Fourth, while outstretching is beneficial, it does not completely eliminate noisy trajectories. Finally, extending the model to handle high-dimensional visual observations would be an intriguing direction for future work.

## 7 ETHICS STATEMENT

Our research introduces a novel problem setting in offline reinforcement learning and hierarchical planning, where we extend insufficient training datasets to solve complex tasks. This involves creating longer trajectories, enabling the training of planners on these enhanced datasets. However, this advancement raises crucial ethical considerations, including biases in decision-making, data privacy concerns, and job displacement risks due to automation. It is vital to pursue this technology responsibly, ensuring it benefits all and addresses social inequalities. Collaboration among researchers, policymakers, and industry stakeholders is essential to align these developments with societal values and promote inclusivity.

## 8 REPRODUCIBILITY STATEMENT

To ensure the reproducibility of our experimental results, all necessary resources will be made publicly available upon acceptance. The implementation details and pseudocode for replicating key findings are presented in Appendix A.

## ACKNOWLEDGEMENT

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

APPENDIX

# A IMPLEMENTATION DETAILS

In this section, we describe the architecture and the hyperparameters used for our experiments.

- We build our code on the Decision Diffuser Ajay et al. (2022). We use a similar architecture for the temporal U-Net.
- We represent the level embeddings with a 2-layered MLP with a one-hot level encoding input. We condition the diffuser on the level embedding to generate multiscale trajectories. For training, we sample different levels and the level determines the resolution of the sampled trajectories.
- Following Diffstitch Li et al. (2024), we use MOPO Yu et al. (2020) for the inverse dynamic and reward models.
- For the stitcher model, we trained a decision diffuser with a short horizon $H$ (`Maze2D-Large`: 80, `Maze2D-XXLarge`: 80, `Gym-MuJoCo`: 50, `Kitchen`: 20)
- We represent the level classifier $f_\phi^L(l|s_1, s_2)$ with a 3-layered MLP with 256 hidden units and ReLU activations. The classifier trained with samples from multiscale trajectories to predict the corresponding level.

## A.1 MAZE2D DATASET

Figure A.6 shows a visualization of the Maze2D-Large and Maze-XXLarge layouts visualizing short trajectories with different colors indicating the region used to collect those trajectories.

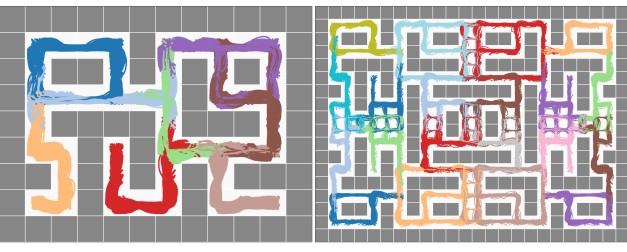

(a) Large Maze        (b) XXLarge Maze

**Figure A.6: Maze2d Maps with visualized short trajectories. (a) Large Maze:** The PointMaze2D large maze environment, where the optimal trajectory from the top-left corner to the bottom-right corner takes approximately 500 steps. **(b) XXLarge Maze:** A newly introduced maze that is twice as long in both dimensions, resulting in a maze that is four times larger than the Large Maze. Consequently, navigating between the two most distant states requires approximately 1000 steps for the PD controller.

## A.2 PROGRESSIVE TRAJECTORY EXTENSION (PTE)

In this section, we first provide the pseudocode of our Progressive Trajectory Extension (PTE) process in algorithm A.2. As discussed earlier, our PTE method allows flexible input datasets, thus enabling different stitchin strategies. In algorithm A.3, we highlighted the process of linear PTE, and the exponetial PTE is depicted in algorithm A.4. Table A.4 shows a comparison between exponential PTE and Linear PTE in terms of trajectory length.

---

**Algorithm A.1** Segmenting and sampling for stitching

---

1: **Input**: Trajectory $\tau = \{s_t, a_t, r_t\}_{t=1}^{T}$
2: **Output**: specific state $s_i$

3: Partition $\tau$ into $K$ non-overlapping sgments and assign probabilities for the segments

4: Sample a segment $b_j = \{s_t, a_t, r_t\}_{t=T_{b_j}}^{T_{b_{j+1}}-1}$

5: Uniformly sample a position $i$ from $\{T_{b_j}, T_{b_j}+1, \ldots, T_{b_{j+1}}-1\}$

6: **Return:** $s_i$

---

**Algorithm A.2** Progressive Trajectory Extension

---

1: **Input:** Trained $p_\theta^{\text{stitcher}}$, Inverse Dynamic Model $f_\theta^a$, Reward Model $f_\theta^r$, Reachability Threshold $\delta$, Source Dataset $\mathcal{S}^r$, Target Dataset $\mathcal{T}^r$, Number of iterations $N$
2: **Output:** Stitched Dataset $D^r$
3: Initialize $D^r \leftarrow \emptyset$
4: **for** $i = 1$ to $N$ **do**
5:     Sample a source trajectory $\tau^{\text{src}} \sim \mathcal{S}^r$ and a batch of candidates $\mathcal{T}_c \subset \mathcal{T}^r$
6:     Obtain $s_t^{\text{src}}$ from $\tau^{\text{src}}$ and $\{s_{c,t''}^{\text{cand}}\}_c$ from each candidate $\tau_c^{\text{cand}} \in \mathcal{T}_c$ using **Algorithm A.1**
7:     Sample a bridge trajectory $\tau^{\text{brg}} \sim p_\theta^{\text{stitcher}}(\tau | s_0 = s_t^{\text{src}})$
8:     Filter out candidate $\tau_c^{\text{cand}}$ and get $\mathcal{T}_{c,\delta} \subset \mathcal{T}_c$ based on:

$$\min_{t'} \|s_{t'}^{\text{brg}} - s_{c,t''}^{\text{cand}}\|^2 > \delta$$

9:     Sort $\mathcal{T}_{c,\delta}$ based on *outstretch score*:

$$\sigma_{\text{outstretch}}(\tau^{\text{src}}, \tau_c^{\text{brg}}, \tau_c^{\text{cand}}) := \frac{\|s_0^{\text{src}} - s_{c,T}^{\text{cand}}\|^2}{t + T - t''}. \tag{3}$$

10:    Randomly sample target trajectory $\tau_c^{\text{tgt}}$ from top $K$ candidates
11:    Re-sample the bridge $\tau^{\text{rebrg}} \sim p_\theta^{\text{stitcher}}(\tau | s_0 = s_t^{\text{src}}, \cdots, s_k = s_{t''}^{\text{tgt}}, \cdots, s_h = s_{t''+h-k}^{\text{tgt}})$
12:    Get $\tau^{\text{new}} = Concat(\tau_{1:t-1}^{\text{src}}, \tau_{0,t'}^{\text{rebrg}}, \tau_{t''+1:T}^{\text{tgt}})$
13:    Update $D^r \leftarrow D^r \cup \tau_{\text{new}}$
14: **end for**
15: **Return:** Extended Dataset $D^r$

---

**Algorithm A.3** Linear PTE

---

1: **Input:** Trained $p_\theta^{\text{stitcher}}$, Inverse Dynamic Model $f_\theta^a$, Reward Model $f_\theta^r$, Reachability Threshold $\delta$, Source Dataset $\mathcal{S}_r = \mathcal{D}_{\text{out}}^{r-1}$, Target Dataset $\mathcal{T}^r = \mathcal{D}^0$, Number of iterations $N$
2: **Output:** Stitched Dataset $D^r$
    Use AlgorithmA.2 with $\mathcal{S}_r = \mathcal{D}_{\text{out}}^{r-1}, \mathcal{T}^r = \mathcal{D}^0$
3: **Return:** Extended Dataset $D^r$

---

**Algorithm A.4** Exponential PTE

---

1: **Input:** Trained $p_\theta^{\text{stitcher}}$, Inverse Dynamic Model $f_\theta^a$, Reward Model $f_\theta^r$, Reachability Threshold $\delta$, Source Dataset $\mathcal{S}^r = \cup_{r'=0}^{r-1} D_{\text{out}}^{r'}$, Target Dataset $\mathcal{T}^r = \cup_{r'=0}^{r-1} D_{\text{out}}^{r'}$, Number of iterations $N$
2: **Output:** Stitched Dataset $D^r$
    Use AlgorithmA.2 with $\mathcal{S}^r = \cup_{r'=0}^{r-1} D_{\text{out}}^{r'}, \mathcal{T}^r = \cup_{r'=0}^{r-1} D_{\text{out}}^{r'}$
3: **Return:** Extended Dataset $D^r$

---

**Table A.4: Comparison of Trajectory Length Statistics Across PTE Rounds in Maze2D-XXLarge**. Exponential PTE shows a more rapid increase in trajectory length, with earlier rounds producing longer maximum trajectories compared to Linear PTE. Linear PTE, on the other hand, demonstrates a steadier, more gradual extension across rounds.

| PTE | Metric | Trajectory Length | | | | | | |
|---|---|---|---|---|---|---|---|---|
| | | Base | Round 1 | Round 2 | Round 3 | Round 4 | Round 5 | Round 6 |
| **Linear** | Mean | 172 | 354 | 493 | 608 | 729 | 849 | 967 |
| | Min | 103 | 219 | 330 | 450 | 563 | 675 | 779 |
| | Max | 343 | 574 | 698 | 838 | 1012 | 1129 | 1321 |
| **Exponential** | Mean | 172 | 355 | 526 | 700 | 981 | N/A | N/A |
| | Min | 103 | 225 | 222 | 264 | 316 | N/A | N/A |
| | Max | 343 | 569 | 839 | 1346 | 1778 | N/A | N/A |

## A.3    PLANNING WITH RECURSIVE HM-DIFFUSER

We present the planning pseudocoe with our proposed recursive HM-Diffuser in algorithm A.5 .

---

**Algorithm A.5** Planning with Recursive HM-Diffuser - Replanning

---

1: **Input:** HM-Diffuser $p_\theta$, Evaluation Environment *env*, Inverse Dynamic $f_\theta^a$, Number of Levels $L$, Jump Count $K = \{k_\ell\}^L$

2: $s_0 = env.\text{init}()$
    $\triangleright$ Reset the environment.

3: done = False
4: **while** not done **do**
5:     **for** $\ell$ in $L, \dots, 1$ **do**
6:         **if** $\ell == L$ **then**
7:             $\tau_g^\ell = \{g_0^\ell, \dots, g_{k_\ell}^\ell\} \leftarrow p_\theta(\boldsymbol{\tau}|\ell, g_0^\ell = s_0)$
    $\triangleright$ Sample a subgoal plan given start.
8:         **else**
9:             $\tau_g^\ell = \{g_0^\ell, \dots, g_{k_\ell}^l\} \leftarrow p_\theta(\boldsymbol{\tau}|\ell, g_0^\ell = s_0, g_{k_\ell}^\ell = g_1^{\ell+1})$
    $\triangleright$ Refine plans given subgoals from one layer above.
10:         **end if**
11:     **end for**
12:     Extract the first two states, $s_0, s_1 = g_0^1$, from the first layer plan $\tau_g^1$
13:     Obtain action $a = f_\theta^a(s_0, s_1)$
14:     Execute action in the envirionment $s, \text{done} = env.step(a)$
15: **end while**

---

**Algorithm A.6** Goal-Conditioned Planning with Recursive HM-Diffuser (w/o Replanning)

1: **Input:** HM-Diffuser $p_\theta$, Evaluation Environment *env*, Inverse Dynamic $f_\theta^a$, Number of Levels $L$, Jump Count $K = \{k_\ell\}^L$, Level Classifier $f_\theta^l$, Maximum number of planning rounds $N_P$

2: $s_0, s_{goal} = env.\text{init}()$
$\qquad\qquad\qquad\qquad\qquad\qquad\qquad\qquad\qquad\qquad\qquad\qquad$ ▷ Reset the environment.

3: done = False
4: done_plan = False
5: $\boldsymbol{\tau} = \{\}$
6: $t_p = 0$
7: **while** not done_plan **do**
8: $\quad$ Obtain level $\ell = f_\theta^l(s_0, s_{goal})$
9: $\quad \boldsymbol{\tau}_g^\ell = \{g_0^\ell, \ldots, g_{k_\ell}^l\} \leftarrow p_\theta(\boldsymbol{\tau}|\ell, g_0^\ell = s_0, g_{k_\ell}^\ell = s_{goal})$
$\qquad\qquad\qquad\qquad\qquad\qquad\qquad\qquad$ ▷ Sample a plan given subgoals from previous layer.
10: $\quad$ Obtain a set of starting states for the next layer $s_0 = \{g_0^\ell, \ldots, g_{k_\ell-1}^l\}$
11: $\quad$ Obtain a set of goal states for the next layer $s_{goal} = \{g_1^\ell, \ldots, g_{k_\ell}^l\}$
12: $\quad$ **if** $\ell == 1$ or $t_p \geq N_P$ **then**
13: $\qquad$ done_plan = True
14: $\quad$ **end if**
15: $\quad \boldsymbol{\tau} \leftarrow \boldsymbol{\tau} \cup \boldsymbol{\tau}_g^\ell$
16: $\quad t_p = t_p + 1$
17: **end while**
18: t = 0
19: **while** not done **do**
20: $\quad$ Obtain action $a_t = f_\theta^a(s_t, \boldsymbol{\tau}[\min(t, len(\boldsymbol{\tau})])$
21: $\quad$ Execute action in the envirinment $s_t$, done $= env.step(a_t)$
22: $\quad t = t + 1$
23: **end while**

**Algorithm A.7** Recursive HM-Diffuser Training

1: **Input:** Recursive HM-Diffuser $p_\theta$, Inverse Dynamic $f_\theta^a$, number of levels $L$, Reward Model $f_\theta^r$, Jumpy Step Schedule $J = \{j^0, \ldots, j^L\}$, Training Dataset $\mathcal{D}$
2: **while** not done **do**
3: $\quad$ Sample a batch of trajectory from dataset $\boldsymbol{\tau} = \{s_t, a_t, r_t\}^{t+h} \sim \mathcal{D}$
4: $\quad$ Sample a level $\ell \sim \text{Unifrom}[0, \ldots, L]$
5: $\quad$ Obtain the sparse trajectory for level $\ell$: $\boldsymbol{\tau}^\ell = (g_0^\ell, \ldots, g_{k_\ell}^\ell)$
6: $\quad$ Train HM-Diffuser with Equation 4
7: $\quad$ Train inverse dynamics $f_\theta^a$
8: $\quad$ Train reward model $f_\theta^r$
9: **end while**

## B    MORE PTE ANALYSIS ON GYM-MUJOCO

In this section, we present additional plots analyzing the averaged return per step from the Gym-MuJoCo dataset after one round of progressive trajectory extension (PTE). As depicted in Figure B.7, there is a noticeable shift from low-value to high-value returns across nearly all datasets following the implementation of one PTE round, except for hopper-medium-replay-v2 and hopper-medium-v2.

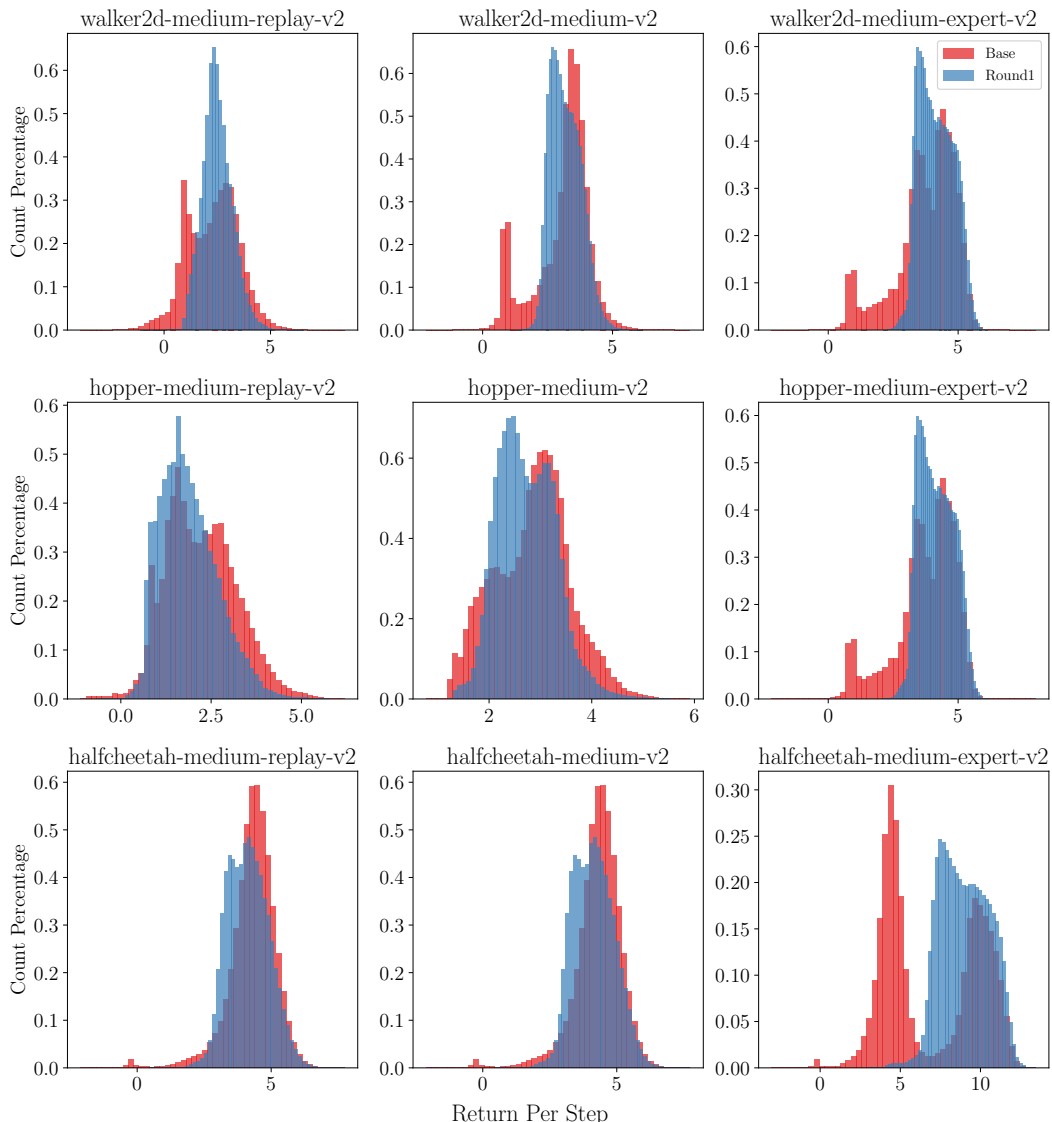

**Figure B.7: Analysis of the progressive trajectory extension process.** In the Gym-MuJoCo task, we observed a noticeable shift towards high values in the distribution of returns per step after one PTE round, suggesting that the stitched trajectories have evolved into trajectories with higher returns.

