# OpenReview forum: "Hierarchical Multiscale Diffuser for Extendable Long-Horizon Planning"
_ICLR.cc/2025/Conference — ICLR 2025 Conference Withdrawn Submission_

### Official Review · Reviewer_VVjy · 2024-11-04

**Soundness:** 2
**Presentation:** 3
**Contribution:** 2
**Rating:** 5
**Confidence:** 3

**Summary:**

The authors propose a hierarchical multi-scale diffusion planner that extends diffusion planning to longer trajectories than they were trained on via a stitching operation. Favorable results are demonstrated on a 2D maze and modified versions of the D4RL/FrankaKitchen benchmarks.

**Strengths:**

- The paper is mostly well-written
- The work addresses a relevant limitation in diffusion planning

**Weaknesses:**

- The results are a difficult to assess as the choice of benchmarks either seem rather trivial (2D maze) or use modified versions of standard benchmarks for which there are no suitable baselines (yet at least). Diffusion policies have previously produced SOTA results on e.g., certain manipulation tasks, results on 2D mazes or your own variants of extended benchmarks against fixed-length diffusion planning is not entirely convincing.

**Questions:**

Shouldn't the extra flexibility in trajectory length of your method be useful in some existing benchmark with more established baselines?

---

### Official Review · Reviewer_cj6K · 2024-11-06

**Soundness:** 2
**Presentation:** 2
**Contribution:** 2
**Rating:** 3
**Confidence:** 3

**Summary:**

This paper introduces the Hierarchical Multiscale Diffuser (HM-Diffuser), a new framework for extending long-horizon planning in reinforcement learning by integrating Progressive Trajectory Extension (PTE) with hierarchical diffusion models. To overcome the constraints of current diffusion-based planners, the authors propose PTE, which iteratively stitches short trajectories into longer sequences, enabling plans that surpass the original data’s horizon. Through experiments in Maze2D, Gym-MuJoCo, and high-dimensional manipulation tasks like FrankaKitchen, HM-Diffuser often performs better than existing models, such as Decision Diffuser (DD) and Hierarchical Diffuser (HD).

**Strengths:**

**Originality**: The paper introduces an approach for long-horizon planning in reinforcement learning (RL) by combining hierarchical diffusion models with Progressive Trajectory Extension (PTE). The PTE mechanism, which stitches shorter trajectories to create longer ones, is an innovative solution to data scarcity in extended planning, expanding RL capabilities in complex, long-term tasks.

**Quality**: The methodology is shown to work, and includes experiments across multiple benchmarks (Maze2D, Gym-MuJoCo, FrankaKitchen) to demonstrate the model’s adaptability to diverse environments, though a more rigorous formulation could enhance the soundness. Comparisons to baselines, such as Decision Diffuser (DD) and Hierarchical Diffuser (HD), show HM-Diffuser’s advantages.

**Clarity**: The structure of the paper is ok, each component is presented. Explanations of diffusion models, hierarchical planning, and the overall framework are clear, making the concepts accessible to those familiar with RL. Visual aids and pseudocode support the narrative, though additional clarifications could further enhance readability.

**Significance**: This work addresses a critical challenge in RL—planning over horizons longer than the training data allows. The individual components seem to fit together well, however, it’s hard to justify the applicability of the combined framework in general, reducing its broader applicability.

**Weaknesses:**

**Overly Ambitious Claims**:  Throughout the paper, the authors motivate that we need models that allow robots to plan over “week- or month-long” horizons based on visual experiences, which is overly ambitious and detracts from credibility. Limiting claims to the demonstrated capabilities would improve reliability.

**Ad-Hoc Components Without Rigorous Justification**: Key components, such as linear and exponential PTE, and APP, appear as ad hoc solutions without clear theoretical or empirical support. This engineered approach may limit generalizability, and a stronger theoretical basis or clarification of assumptions is needed.

**Unclear Scoring and Metrics**: The scores in Tables 1, 2, and 3 lack clear explanation, making it difficult to interpret model performance. Providing definitions and explanations for these metrics would improve result clarity.

**Omission of Training Time and Resource Comparisons**: No information on training times or resource demands is provided. Given the model’s complexity, details on computational efficiency would help assess feasibility for real-world use.

**Lack of Compounding Error Analysis**: The paper claims to tackle one fundamental issue with prior methods, i.e., compounding errors, but does not provide a direct comparison with other models. An explicit analysis would strengthen the validation of this claim.

**Missing Subgoal Visualizations**: Visuals of subgoals generated by the Hierarchical Multiscale Diffuser would clarify the multiscale planning process and show how subgoals contribute to task performance.

**Undefined Contribution of Hierarchical Multiscale Diffuser**: The paper does not isolate the performance improvements introduced by the Hierarchical Multiscale Diffuser (HMD) over PTE alone. Quantifying these contributions would clarify the value of HMD.

**No Ablation Study on Key Parameters**: An ablation study on jump lengths and counts is missing, which would help clarify their impact on model performance across tasks and aid in parameter tuning.

**Reproducibility Limitations**: Although hyperparameters and references are provided, sharing the code (through an anonymized repository) would enhance reproducibility, especially given the model’s complexity.

**Insufficient and Unreferenced Visuals**: Additional visuals and proper referencing, especially for Figures 1 and 2, would improve clarity on hierarchical planning and recursive processes.

**Fixed Segment Sizes Limit Generalizability**: The use of fixed segment sizes in trajectory stitching may limit adaptability to complex tasks like Franka Kitchen, where variable segment sizes may be necessary.

**Lack of Comparison to Existing Stitching Methods**: Existing stitching methods are briefly mentioned as limited (line 177) without specification, making it hard to assess PTE’s novelty. A clear comparison would better contextualize the contributions of PTE.

**Questions:**

Most of my comments on the ‘weaknesses’ section can be treated as questions. Here are some other points:

**Trajectory Information**: Do the trajectories contain only positional information (2D or 3D), or do they also include other state variables, such as velocity or acceleration? How does this impact the model’s generalizability across different environments?

**Outstretching and Bridge Trajectory Sampling**: Euclidean distance is used for outstretching and bridge trajectory sampling. Given that this may not capture feasible paths in complex environments, e.g., for constrained robotic (manipulation) tasks, how is the distance threshold determined, and do you adjust it for different environments? What mechanisms, if any, are in place to account for constraints, or to ensure that the stitched trajectory remains feasible?

**Trajectory Feasibility Checks**: Are there any feasibility checks for the complete trajectory generated by PTE? If a stitched trajectory is infeasible, what measures are in place to detect and handle this?

**Hierarchical Diffuser and Position-Based Goals**: It appears the Hierarchical Diffuser primarily generates subgoals based on position. Could it be adapted to environments where position alone may not represent effective goals, or where additional context (e.g., velocity or object interactions) is needed?

**Realism and Diversity of Generated Trajectories**: The example maze trajectories in Figure 3 appear repetitive. Does it play a role in the algorithm performance?

**Application to High-Dimensional Visual Data**: In the discussion of future work, you mention visual observations. How do you envision adapting distance-based metrics like Euclidean distance in pixel-based environments, where state representation is more complex?

---

### Official Review · Reviewer_QCtk · 2024-11-06

**Soundness:** 3
**Presentation:** 3
**Contribution:** 3
**Rating:** 6
**Confidence:** 3

**Summary:**

The paper presents an approach for efficient long-horizon planning that addresses the challenge of planning over horizons longer than those encountered during training. The present their Hierarchical Multiscale Diffuser (HM-Diffuser) framework, that consists of two main steps, the Progressive Trajectory Extension (PTE) and the hierarchical multiscale planner (HMD). PTE stitches together short trajectories to generate longer ones, while HMD trains on these extended datasets to improve its long-horizon planning capabilities. The paper further introduces several improvements to HMD, including Adaptive Plan Pondering and a recursive version of HMD, which uses a single model to handle multiple temporal scales. The authors present experiments on a set of planning tasks that demonstrate how HMD can generate long-horizon plans that extend beyond the original examples trajectories and that out perform decision diffuser (DD) and hierarchical diffuser (HD) approaches.

**Strengths:**

In terms of originality, both the HM-diffuser and the PTE methods are novel. In addition, the extension of the benchmarks with the long-horizon versions can be useful for the community.
The authors present the problem clearly and support their claims with experimental results and evaluations in a set of (extended) benchmarks, though the evaluation of the results can benefit from a clearer presentation (see below).
The paper is well-organized and the related work is well structured. The paper is easy to follow and the authors provide a good level of detail for the implementation in the paper and in the appendix.
The work addresses a significant challenge and the evaluation against existing and extended benchmarks adds to the potential impact of the proposed approach.

**Weaknesses:**

While the results for the long-horizon planning for the 2d-maze are well presented and illustrated, the evaluation on the offline rl side can be better presented. It is not clear what longer paths mean for the gym-mujoco examples and for the kicthen task scenarios. What do the performance percentages represent in Table 2? Similarly, what the numbers in table 3 represent in terms of performance in the kitchen task is not clear. A more detailed explanation of the evaluation process an help with the clarity in the results section.

**Questions:**

Typo in line 431, necessarity -> necessity

---

### Official Review · Reviewer_npsz · 2024-11-17

**Soundness:** 2
**Presentation:** 3
**Contribution:** 3
**Rating:** 6
**Confidence:** 4

**Summary:**

The paper presents the Hierarchical Multiscale Diffuser (HM-Diffuser), an approach for  long-horizon planning, leveraging diffusion-based planning to handle planning tasks with timelines longer than those found in training data. The authors propose to break down the planning process into two main stages. First, they introduce Progressive Trajectory Extension (PTE), a method for combining shorter trajectories into longer datasets. Then, they train the HM-Diffuser on these extended datasets, maintaining computational efficiency while enhancing long-term planning. HM-Diffuser’s hierarchical design enables it to generate subgoals at various temporal scales, facilitating a top-down approach that bridges high-level objectives and immediate actions. Experiments show that this combined approach successfully produces plans that exceed the initial trajectory lengths, demonstrating effective long-horizon planning capabilities.

**Strengths:**

- Paper is well written, clear and well-structured. The motivation and the idea of the paper is clear. The approach is clearly stated and all relevant background is introduced well. The method explanation is accompanied by appropriate visualisation figures that help to understand.
- The proposed method seems reasonable and well-thought-of, with clear description and theoretical backing.
- The proposed method achieves good performance on target experiments.
- The approach is tested in several simulation experiments.

**Weaknesses:**

- The choice of subgoals seems a bit arbitrary and there should be more analysis and ablation studies in these. It is not clear fully how this coise for the segment length and extensions work in different problems. E.g. in XXL Maze they seem to work well. It is not clear for the other domains.
- The name for section 5.3.3. is not informative.
- PD controller used here is not explained
- The improvements on Franka Kitchen Task in table 3 seem to be marginal.

**Questions:**

- Can you provide more intution on choosing subgoals? In which environments should it work better?
- What ensures the feasibility of trajectories?
- Can you provide more information about PD controller?
- In Figure 4 the x axis as it is seems a bit unsual and hard to interpret. Perhaps you could use stacked barplot?

---

### Note · Authors · 2024-11-25

**Comment:**

Thank you for your valuable feedback and the time you dedicated to reviewing our paper. We greatly appreciate your constructive comments and suggestions. After careful consideration, we have decided to withdraw this submission and revise it further before resubmitting.

**Withdrawal Confirmation:**

I have read and agree with the venue's withdrawal policy on behalf of myself and my co-authors.